# The Candidate IBD Risk Gene *CCNY* Is Dispensable for Intestinal Epithelial Homeostasis

**DOI:** 10.3390/cells10092330

**Published:** 2021-09-06

**Authors:** Andrea Molinas, Stéphanie Heil, Stefan Koch

**Affiliations:** 1Department of Biomedical and Clinical Sciences (BKV), Linköping University, s-581 85 Linköping, Sweden; andrea.molinas@liu.se (A.M.); stephanie.heil@liu.se (S.H.); 2Wallenberg Centre for Molecular Medicine (WCMM), Linköping University, s-581 85 Linköping, Sweden

**Keywords:** Wnt signaling, inflammatory bowel diseases, colitis, CDK14

## Abstract

The *CCNY* gene, which encodes cyclin Y, has been implicated in the pathogenesis of inflammatory bowel disease (IBD). Cyclin Y promotes Wnt/β-catenin signaling and autophagy, which are critical for intestinal epithelial cell (IEC) homeostasis, and may thereby contribute to wound repair in colitis. However, whether cyclin Y has an essential function in IECs is unknown. We, therefore, investigated the epithelial injury response and mucosal regeneration in mice with conditional knock-out of *Ccny* in the intestinal epithelium. We observed that *Ccny*-deficient mice did not exhibit any differences in cell proliferation and disease activity compared to wild-type littermates in the dextran sulfate sodium (DSS) colitis model. Complementary in vitro experiments showed that loss of *CCNY* in model IECs did not affect Wnt signaling, cell proliferation, or autophagy. Additionally, we observed that expression of the cyclin-Y-associated cyclin-dependent kinase (CDK) 14 is exceedingly low specifically in IEC. Collectively, these results suggest that cyclin Y does not contribute to intestinal epithelial homeostasis, possibly due to low levels of specific CDKs in these cells. Thus, it is unlikely that *CCNY* mutations are causatively involved in IBD pathogenesis.

## 1. Introduction

Inflammatory bowel disease (IBD) is a common disorder that involves chronic, recurring inflammation of the intestinal epithelium with a considerable impact on the patients’ quality of life. IBD has a high prevalence in industrialized countries and an increasing incidence in developing countries, and its etiology remains largely elusive [1]. Genome-wide association studies have identified several genetic risk loci shared between the main disease entities, ulcerative colitis (UC) and Crohn’s disease (CD) [2], that are thought to contribute to IBD pathogenesis.

Several single nucleotide polymorphisms identified in these studies have been mapped to the *CCNY* gene encoding the cell cycle regulator cyclin Y [3,4,5,6,7], whose biological functions are incompletely understood. *Ccny*-deficient mice display altered lipid metabolism and lower body weight, while double knock-out mice for *Ccny*, and its less ubiquitously expressed homolog *Ccnyl1* (cyclin Y-like 1), are embryonic lethal [8,9,10]. These observations suggest that cyclin Y is essential for normal development and physiology.

Cyclin Y and cyclin Y-like 1 are potent activators of the homeostatic Wnt/β-catenin signaling pathway [11,12]. Mechanistically, a cyclin Y/cyclin-dependent kinase (CDK) 14 complex activates the Wnt pathway through phosphorylation of Wnt coreceptor LRP6, which is thereby primed for incoming Wnt ligands. Wnt signaling peaks during G2 phase and mitosis when cyclin Y protein levels peak, suggesting cell-cycle-dependent Wnt pathway activation [11]. Wnt/β-catenin signaling is critically involved in maintaining the intestinal epithelial stem cell pool, and, accordingly, activation of Wnt signaling is thought to contribute to mucosal regeneration in colitis by increasing epithelial proliferation [13,14]. Consistently, it was recently shown that intestinal epithelial cell (IEC)-specific loss of *Lrp6* increases the susceptibility to experimental colitis, and reduces enteroid growth ex vivo [15]. Cyclin Y has also been implicated in the regulation of autophagy. Dohmen and colleagues [16] reported that cyclin Y is a substrate of AMP-activated protein kinase (AMPK), and that AMPK-dependent cyclin Y phosphorylation promotes autophagy via its association with CDK16. Mutations in autophagy-related genes are frequently observed in IBD, and defective autophagy is considered a key pathogenic event in CD [17].

Given these observations, we hypothesized that cyclin Y is involved in intestinal epithelial cell homeostasis and wound repair during colitis. Contrary to these expectations, we found that IEC-specific *Ccny* knock-out mice were indistinguishable from wild-type littermates in terms of crypt architecture, epithelial proliferation, and mucosal regeneration following experimental colitis. Complementary in vitro experiments further indicated that cyclin Y does not regulate Wnt signaling or autophagy in IEC, possibly due to low expression of relevant CDKs. Collectively, our data suggest that *CCNY* is likely not involved in intestinal epithelial homeostasis and IBD pathogenesis.

## 2. Materials and Methods

### 2.1. Mice

*Ccny* floxed mice in a C57Bl/6 background [8] were kindly provided by Dr. Xueliang Zhu (Shanghai Institutes for Biological Sciences, Shanghai, China). Intestinal epithelial-cell-specific knock-out mice (*Ccny*^Δ^^IEC^) were generated by mating with Villin-cre mice (B6.Cg-Tg(Vil1-cre)997Gum/J, Jackson Laboratories stock # 004586). Mice were housed in an SPF facility in a standard day–night cycle with unrestricted access to food and water. Genotypes were determined by PCR for the *Villin* transgene and *Ccny*.

### 2.2. Dextran Sulfate Sodium (DSS) Colitis

All animal experiments were reviewed and approved by the regional ethics committee (Linköpings djurförsöksetiska nämnd, Linköping, Sweden, ID1037), and performed according to national and institutional guidelines. Colitis was induced by adding 3–3.5% *w*/*v* DSS (TdB Labs, Uppsala, Sweden; DB001, molecular weight 40,000 Da) to the drinking water, which was provided ad libitum for five days. For recovery, mice were switched back to regular drinking water for an additional five days. Mice of both sexes were used at 8–10 weeks of age, and assigned randomly to the colitis or control groups. Colitis was monitored by daily recording of a modified clinical disease activity index (DAI) [18,19]. In brief, the DAI is a composite score that assesses body weight change, stool composition, and presence of blood in the stool, as determined by a Hemoccult guaiac test (Beckman Coulter, Solna, Sweden). The individual parameters are scored on a scale of 0–4, from lowest to highest severity, added up, and divided by 3. No DAI was determined if a stool pellet could not be collected after ~20 min of observation. For this reason, body weight changes are shown separately as a surrogate marker of disease activity. At the experimental or ethical endpoint, mice were anesthetized by isoflurane inhalation, and killed by cervical dislocation. The intestine (from distal ileum to rectum, excluding the cecum) was then removed and prepared as Swiss rolls, as described [20]. Tissue samples were stored in Tissue-Tek OCT (Sakura Finetek, Alphen aan den Rijn, The Netherlands) until further processing.

### 2.3. Cell Culture and siRNA Transfection

HCT116, HEK293T, SW48, and PC-3 cells were obtained from the German Collection of Micro-organisms and Cell Cultures (DSMZ, Braunschweig, Germany), and were cultured in Dulbecco’s modified Eagle medium (DMEM) (Thermo Fisher Scientific, Göteborg, Sweden) containing 10% heat-inactivated fetal calf serum (FCS) (Thermo Fisher Scientific, Göteborg, Sweden) and 1% penicillin/streptomycin (Thermo Fisher Scientific, Göteborg, Sweden). Caco-2 cells were from the European Collection of Authenticated Cell Cultures (ECACC, Salisbury, UK), and maintained in DMEM with 15% FCS, 2 mM L-glutamine, and antibiotics. Wnt3a-producing and control L cells were from the American Type Culture Collection (ATCC, Manassas, VA, USA), and conditioned media were produced as described in the supplier’s protocol. Low-passage cells from confirmed mycoplasma-free frozen stocks, as determined by analytical qPCR (Eurofins Genomics, Ebersberg, Germany), were used for all experiments. Cells were maintained in a humidified incubator at 37 °C/5% CO_2_ and passaged every 2–3 days with trypsin when the cells reached ~80% confluency. For the MTT proliferation assay and the luciferase reporter assay, HCT116 or HEK293T cells were seeded in tissue-culture-treated 96-well plates at a density of 5 × 10^4^ cells per well (MTT assay) or 3 × 10^4^ cells per well (luciferase reporter assay), and transfected after 24 h with *CCNY* siRNAs (Silencer Select Pre-designed siRNA s47718 and s47719, Thermo Fisher Scientific, Göteborg, Sweden) or scrambled control at a final concentration of 20 nM, using Lipofectamine 2000 transfection reagent (Thermo Fisher Scientific, Göteborg, Sweden) according to the manufacturer’s instructions. In brief, 25 μL of a 20 μM stock dilution of the respective siRNA in Opti-MEM was mixed with 1 µL Lipofectamine 2000 transfection reagent diluted in 25 µL Opti-MEM. The transfection mixture was incubated for 15 min at room temperature and added dropwise to the cells. Then, cells were incubated for at least 24 h. For the autophagy experiments, HCT116 cells were seeded in tissue-culture-treated 24-well plates at a density of 1.5 × 10^5^ cells per well, and transfected after 24 h with *CCNY* siRNAs at a final concentration of 20 nM, using Screenfect (ScreenFect GmbH, Eggenstein-Leopoldshafen, Germany) transfection reagent according to the manufacturer’s instructions. In brief, siRNA suspended in 40 µL dilution buffer was mixed with 1 µL Screenfect transfection reagent in 40 µL dilution buffer. The transfection mixture was incubated for 20 min at room temperature to allow complex formation, and then added dropwise to the cells. Cells were incubated for at least 24 h, and then switched to complete growth media or glucose starvation media (DMEM without glucose, supplemented with 10% FBS) with 100 nM Bafilomycin A1 (MedChemExpress, Monmouth Junction, NJ, USA) for 16 h.

### 2.4. Immunoblotting

Cells were seeded in tissue-culture-treated 24-well plates at a density of 1.5 × 10^5^ cells per well 24 h prior to treatments, as described in “Cell culture and siRNA transfection”. Whole-cell extracts were prepared by lysis in 1% NP-40 lysis buffer containing protease and phosphatase inhibitors (Thermo Fisher Scientific, Göteborg, Sweden). Cell lysates were boiled in 4× Laemmli loading buffer, and proteins were resolved on SDS-PAGE polyacrylamide gels, followed by transfer to nitrocellulose membranes (Bio-Rad, Hercules, CA, USA). Membranes were incubated with the following antibodies: anti-cyclin Y (rabbit, 1:5000, 18042-1-AP, Proteintech, Rosemont, IL, USA; or rabbit, 1:200, HPA036290, Sigma-Aldrich, Darmstadt, Germany), anti-CDK14 (mouse, 1:200, sc-376366, Santa Cruz Biotechnology, Dallas, TX, USA), anti-LC3B (rabbit, 1:1000, 2775, Cell Signaling Technology, Danvers, MA, USA), anti-LRP6 Sp1490 (rabbit, 1:1000, 2568T, Cell Signaling Technology, Danvers, MA, USA), anti-LRP6 (mouse, 1:1000, sc-25317, Santa Cruz Biotechnology, Dallas, TX, USA), anti-alpha-tubulin (mouse, 1:5000, NB100-690, Novus Biologicals, Centennial, CO, USA), anti-transferrin receptor/CD71 (mouse, 1:1000, sc-65882, Santa Cruz Biotechnology, Dallas, TX, USA); followed by anti-mouse or anti-rabbit NIR fluorophore-conjugated secondary antibodies (1:20,000, LI-COR Biotechnology, Lincoln, NE, USA). Membranes were imaged on LICOR CLx system and Western blot quantification was performed using LICOR Image Studio 3.2 software. Where required, images were adjusted linearly in ImageJ2/Fiji (NIH, Bethesda, MD, USA). Original blot images can be found in the Appendix A.

### 2.5. Immunostaining

HCT116 and HEK293T cells were seeded in Ibidi optical 24-well plates (Ibidi GmbH, Gräfelfing, Germany) at a density of 1.5 × 10^5^ cells per well and grown for 24 h prior to treatments, as described above. Cells were fixed with 4% PFA and permeabilized with 0.25% Triton X-100. Unspecific binding was blocked with 10% donkey serum. Cells were incubated with the following antibodies: anti-LC3B (1:200, 2775, Cell Signaling Technology, Danvers, MA, USA) and anti-rabbit conjugated to DyLight 555 (1:500, SA5-10039, Thermo Fisher Scientific, Göteborg, Sweden), followed by counterstaining with NucBlue (Thermo Fisher Scientific, Göteborg, Sweden) and mounting with ProLong Glass (Thermo Fisher Scientific, Göteborg, Sweden). Images were obtained on an inverted wide-field Leica DMi8 microscope (Leica Mycrosystems, Wetzlar, Germany) equipped with a Hamamatsu Orca Flash 4 sCMOS LT camera (Hamamatsu, Hamamatsu City, Japan) using a 63×/1.2 objective.

Tissue samples were sectioned at 7 µm thickness on a Leica CM1950 cryostat (Triolab, Sweden). Sections were fixed using 4% Formalin (Histolab, Sweden) and sections were incubated with blocking buffer (PBS with 2% BSA, 0.1% Tween 20) in a humid chamber for 1 h. Slides were washed in PBS and incubated overnight at 4 °C with primary antibody diluted in blocking buffer (Ki67, rabbit, 1:200, GTX166667, GeneTex, Irvine, CA, USA; CCNY, rabbit, 1:1000, 18042-1-AP, Proteintech; CDK14, mouse, 1:100, sc-376366, Santa Cruz Biotechnology, Dallas, TX, USA; Vimentin, rabbit, 1:500, 5741, Cell Signaling Technology, Danvers, MA, USA). Slides were rinsed and incubated with 1:200 secondary antibody for 1 h at room temperature in a humid chamber. Slides were rinsed and mounted in Prolong Gold Antifade with DAPI or NucBlue (Invitrogen). For Ki67 staining analysis, images from the distal colon were acquired using a Nikon E800 fluorescence microscope controlled through NIS elements software (Nikon Instruments Inc, Tokyo, Japan), and evaluated in a blinded fashion. Two sections per animal were stained in each experiment.

### 2.6. MTT Assay

HCT116 cells were seeded in tissue-culture-treated 96-well plates at a density of 2 × 10^4^ cells per well and transfected with siRNAs at a final concentration of 20 nM, using Lipofectamine 2000 transfection reagent (Thermo Fisher Scientific, Göteborg, Sweden), as described in “Cell culture and siRNA transfection”. Cells were treated after 3 to 6 h with Wnt agonist Wnt3a or control conditioned media (1:5 dilution in complete media). Cell proliferation was assessed at different time points using the MTT assay. Absorbance was measured on a Spark 10 M plate reader (Tecan, Männedorf, Switzerland), using a test wavelength of 570 nm and a reference wavelength of 630 nm.

### 2.7. Luciferase Reporter Assay

HCT116 and HEK293T cells were seeded in tissue-culture-treated 96-well plates at a density of 3 × 10^4^ cells per well and transfected with siRNAs at a final concentration of 10 or 20 nM, using Lipofectamine 2000 transfection reagent (Thermo Fisher Scientific, Göteborg, Sweden), as described in “Cell culture and siRNA transfection”. After 24 h, cells were transfected with a β-catenin/TCF luciferase reporter (Super 8× TOPflash; a gift from Randall Moon, Addgene plasmid #12456) and a Renilla control plasmid (pIS1; a gift from David Bartel, Addgene plasmid #12179), using Lipofectamine 2000. Cells were treated after 3 to 6 h with Wnt agonist Wnt3a or controls. After 24 h, cells were lysed using a homemade dual luciferase assay, following a protocol described in [21]. Luminescence was measured using the Spark 10 M plate reader (Tecan), and TOPflash values were normalized to Renilla activity.

### 2.8. Analysis of Public Datasets

The following preprocessed RNA-seq datasets were used: SCP259 [22] (*n* = 365,492 cells from 18 UC patients and 12 healthy controls. Accessed through the Single Cell Portal at https://singlecell.broadinstitute.org/single_cell; date of accession: 16 November 2020), GSE117993 [23] (*n* = 55 non-IBD controls, 43 UC, 32 colonic CD, 60 ileal CD. Version 22 May 2019; accessed through GREIN at http://www.ilincs.org/apps/grein/?gse=; date of accession: 18 November 2020), E-MTAB-2706 [24] (*n* = 675 cell lines. Version October 18 2016; accessed through the EMBL-EBI Expression Atlas at https://www.ebi.ac.uk/gxa/home; date of accession: 18 November 2020). Single-cell RNA-seq data for the epithelial tSNE were down-sampled to 20,000 cells, and visualized in the Single Cell Portal. All other data were further processed and visualized in R 4.0.2 [25].

### 2.9. Statistics and Data Analysis

Bar and line graphs in all figures depict mean ± standard deviation. Statistical tests are reported in the figure legends, and were performed in Prism 8 (Graphpad Software, San Diego, CA, USA) or R. Results from *Ccny*^ΔIEC^ mice were pooled from several experiments performed under identical conditions, and *n* indicates the total number of mice per group at the beginning of the experiment.

## 3. Results

### 3.1. Cyclin Y Is Expressed in Crypt Base Intestinal Epithelial Cells

*CCNY* is widely expressed [12,26], but its specific function in intestinal epithelial cells (IEC) has not been investigated so far. We, therefore, first determined its expression and subcellular localization in the colonic mucosa. Immunostaining of mouse colon sections detected cyclin Y protein primarily in epithelial cells (Figure 1a). Here, cyclin Y localized to the apical and lateral plasma membrane, consistent with earlier observations in epithelial cells lines [11,12]. To determine whether *CCNY* expression is restricted to specific IEC subpopulations, we next analyzed public gene expression datasets from IBD patients and healthy controls [22,23]. We observed that *CCNY* expression peaked in stem and cycling transit-amplifying cells, whereas levels of its functional homolog *CCNYL1* were highest in mature enterocytes (Figure 1b,c and Appendix A). *CCNY* and *CCNYL1* expression increased slightly, albeit statistically significantly, in UC and CD mucosa, respectively (Appendix A). Curiously, although the expression pattern of *CDK16* overlapped with that of *CCNY*, the gene expression data suggested exceedingly low levels of the cyclin Y effector kinase *CDK14* in the intestine (Figure 1d,e and Appendix A). To explore this observation further, we tested the levels of CDK14 protein in various cell types by immunoblotting (Appendix A). We observed that, in contrast to other cells included in this assay, CDK14 was not detectable in the colorectal cancer cell lines at our disposal. Moreover, immunostaining of mouse colon tissue revealed that CDK14 was restricted to the subepithelial stroma (Figure 1d). We additionally analyzed public expression data from a large panel of cancer cell lines of different tissue origins [24]. Whereas expression levels of *CCNY*, *CCNYL1*, and *CDK16* were broadly comparable across all cell types, the average level of *CDK14* was more than an order of magnitude lower in colon cancer cells compared to any other tissue included in this analysis (Appendix A). Collectively, these results suggest that cyclin Y and CDK16 localize preferentially to crypt base colonic IEC, and that CDK14 levels are low specifically in the large intestine.

### 3.2. Loss of Ccny Does Not Exacerbate Experimental Colitis or Impair Injury Repair in Mice

Because *CCNY* has been implicated in IBD pathogenesis [3,4,5], and our analyses supported a role in intestinal epithelial stem cells, we investigated the function of cyclin Y in IEC in vivo. To this end, we generated IEC-specific conditional knock-out mice by mating *Ccny* floxed with Villin-cre mice (*Vil1*^+/tg^; *Ccny*^f^^/^^f^, hereafter *Ccny*^Δ^^IEC^). *Ccny*^Δ^^IEC^ mice were born at the expected Mendelian ratios and, in contrast to *Ccny* global knock-out animals [8], displayed no overt phenotype. Moreover, visual inspection of tissue sections from the distal ileum and colon did not reveal obvious differences in intestinal histomorphology in transgenic mice, consistent with recent observations from Lrp6^Δ^^IEC^ mice [15]. Immunoblot analysis of colonic mucosal scrapings showed a near-complete loss of cyclin Y protein in *Ccny*^Δ^^IEC^ mice (Figure 2a), further supporting that almost all cyclin Y in the mucosa localizes to the intestinal epithelium. It is known that compensatory mechanisms prevent spontaneous colitis in genetically susceptible animals with intestinal epithelial dysfunction [27]. We, therefore, challenged *Ccny*^Δ^^IEC^ mice with the chemical irritant dextran sulfate sodium (DSS), which causes epithelial erosion and subsequent colitis (Appendix A). We observed no difference in body weight loss and clinical disease activity in *Ccny*^Δ^^IEC^ mice compared to heterozygous and wild-type littermates (Figure 2b,c). Next, we investigated the intestinal injury repair of *Ccny*^Δ^^IEC^ mice by removing DSS after 5 days, which activates a Wnt-dependent mucosal recovery program [28,29]. After 5 days of recovery, *Ccny*^Δ^^IEC^ mice were indistinguishable from littermate controls in terms of weight loss and disease activity (Figure 2d,e), and this was also the case when the results were further stratified by sex (Appendix A). Finally, several mice had to be killed before the end of the experiment because they reached an ethical endpoint, and Kaplan–Meier survival analysis revealed no significant difference between the groups (Appendix A). We conclude that cyclin Y is dispensable for intestinal epithelial homeostasis and wound repair in experimental colitis.

### 3.3. Loss of Cyclin Y Does Not Affect Wnt Signaling in Colorectal Cancer Cells

Cyclin Y activates the Wnt/β-catenin signaling pathway via phospho-priming of the Wnt coreceptor LRP6 in various cells and tissues [9,11,12]. Since Wnt/β-catenin signaling is essential for injury repair in the gut [13,14,30], we investigated if cyclin Y activates the Wnt pathway in model cell lines. Wnt signaling was monitored using a β-catenin/TCF luciferase reporter (TOPflash) in HCT116 colorectal cancer cells, which retain Wnt responsiveness despite activating pathway mutations [31], as well as 293T kidney epithelial cells that served as a positive control (see [11,12]) (Figure 3a,b). Cyclin Y loss-of-function was performed by RNA interference using two independent *CCNY* siRNAs with an average knock-down efficiency of 50–70% compared to scrambled control siRNA (Appendix A). Treatment with Wnt agonist Wnt3a activated the TOPflash reporter in both cell lines, but loss of *CCNY* reduced pathway activity only in 293T cells (Figure 3a). Of note, *CCNY* siRNA #2 was consistently more efficient in depleting cyclin Y, and had a stronger effect in this and the following assays. Consistent with the TOPflash data, immunoblotting revealed that *CCNY* loss-of-function did not impair Wnt-induced LRP6 phosphorylation in HCT116 (Figure 3c). Collectively, the results suggest that cyclin Y is not required for the activation of Wnt signaling in IECs.

### 3.4. Loss of Cyclin Y Does Not Affect Cell Proliferation in Intestinal Epithelial Cells

Since cell proliferation is essential for intestinal epithelial homeostasis and is controlled primarily by Wnt signaling [30], we additionally investigated if cyclin Y regulates IEC proliferation. Cell proliferation was assessed over a 96-h period in HCT116 cells using the MTT tetrazolium dye assay upon cyclin Y loss-of-function by RNA interference, as before. Wnt pathway activity was modulated by treatment with control conditioned media (Figure 4a) or media conditioned with Wnt3a (Figure 4b). We observed that loss of cyclin Y did not affect cell proliferation over the 4-day observation period. Consistently, immunostaining for proliferation marker Ki67 following *CCNY* depletion did not reveal any apparent differences in the number of positive cells compared to control siRNA (Figure 4c). In good agreement with these findings, *Ccny*^Δ^^IEC^ mice showed no difference in colonic crypt length or the extent of the Ki67-positive, proliferative crypt cell compartment compared to littermate controls during normal homeostasis, acute experimental colitis, and following recovery from DSS colitis (Figure 4d–g). We, thus, conclude that cyclin Y is dispensable for cell proliferation in IECs.

### 3.5. Cyclin Y Is Not Required for Autophagy in Stressed Intestinal Epithelial Cells

Recent studies have shown that cyclin Y is also involved in the regulation of autophagy in various cell types via its association with effector kinase CDK16 [16]. To explore if cyclin Y regulates autophagy in IECs as well, we monitored autophagy in HCT116 cells upon cyclin Y knock-down by RNA interference. Cells were treated with the lysosomal protease inhibitor Bafilomycin A1 in complete media or glucose starvation media for 16 h. Autophagosome formation was measured by immunofluorescence analysis of endogenous microtubule-associated protein light chain 3 (LC3) (Figure 5a,b). In striking contrast to the findings by Dohmen and colleagues [16], we observed that loss of *CCNY* increased autophagosome formation in complete media, but had essentially no effect after glucose starvation. Based on these data, we conclude that cyclin Y is not essential for autophagy during cell stress in IEC.

## 4. Discussion

Our results indicate that the candidate IBD risk gene *CCNY* is dispensable for intestinal epithelial homeostasis. The DSS colitis model showed that *Ccny* mutant mice do not present differences in epithelial histomorphology, proliferation, and disease activity compared to wild-type mice. Consistently, functional assays revealed that loss of cyclin Y does not affect Wnt signaling and cell proliferation in the model IEC cell line HCT116. The analysis of autophagosome formation indicated that cyclin Y is not required for autophagy in stressed intestinal epithelial cells. Collectively, these findings support that cyclin Y is not essential for intestinal epithelial homeostasis. It should be noted, however, that the conclusions from in vitro studies are currently limited to HCT116 cells, since other IEC cell lines we tested (Caco-2, SW48) displayed limited Wnt responsiveness and *CCNY* knock-down efficiency (not shown).

It has been shown that cyclin Y activates Wnt/β-catenin signaling in various cell types, including epithelial cells [9,11,32]. In contrast, our data from HCT116 cells suggest that cyclin Y is uncoupled from LRP6 phosphorylation and Wnt signaling in intestinal epithelia, a phenomenon that has also been observed in hepatocellular carcinoma cells [33]. Importantly, although HCT116 are colorectal cancer cells that harbor an activating mutation in β-catenin, they respond to Wnt ligands by LRP6 activation and increased proliferation [31], making them a suitable model for studying cyclin Y biology. How the cell-type specificity of cyclin-Y-dependent Wnt receptor phosphorylation is achieved is currently unclear, but it may involve the differential expression of cyclin-Y-associated effector kinases. Earlier studies suggested that CDK14 is the main regulator of cyclin-Y-dependent Wnt signaling [11,34], and we observed exceedingly low levels of CDK14 specifically in the intestinal epithelium [22,23]. CDK14 has been shown to interact with other cyclins as well, such as cyclin B to activate the Wnt pathway in breast cancer [35], cyclin D3 to regulate cell cycle progression and proliferation in mammalian cells [36], and the CDK7/cyclin H complex to regulate proliferation and migration in esophageal cancer [37]. It should be noted that previous studies suggested functional redundancy among the human PFTAIRE/PCTAIRE family kinases (CDK 14–18) [11,38]. Nonetheless, cell-type specificity could conceivably be achieved by differential temporal or spatial expression of *CCNY* and its associated CDKs [38]. Moreover, it is possible that other proteins required for cyclin-Y-dependent phospho-priming of LRP6, such as caprin-2 [39], exhibit compartmentalized expression as well. Cell proliferation in intestinal epithelial cells is driven by Wnt signaling [30]. Thus, our observation that cyclin Y is not essential for IEC proliferation in vitro and in vivo may be explained by the apparent uncoupling of cyclin Y from Wnt pathway regulation in the gut. It was recently reported that mice with IEC-specific deletion of *Lrp6* are more susceptible to DSS colitis, and that enteroids derived from these animals exhibit reduced proliferation ex vivo [15]. In light of these findings, it would be of considerable interest to further explore the control of Wnt receptor activation in intestinal epithelia.

Cyclin Y has also been implicated in the regulation of autophagy in different cell types via its association with effector kinase CDK16 [16]. However, we were unable to recapitulate a key finding of this earlier study, namely that loss of *CCNY* reduces autophagosome formation in starved cells, using intestinal epithelial cells lines. It is possible that this discrepancy can be explained by cell type-specific effects, as discussed above. An alternative explanation is experimental variation. We initially induced nutrient starvation with EBSS medium as in the Dohmen et al. study [16], but observed considerable cell death in HCT116 cells, which precluded us from drawing any meaningful conclusions. We, therefore, monitored autophagy in IECs under glucose starvation instead of EBSS-mediated starvation, as was previously done in a study on autophagy in various colorectal cancer cell lines [40]. Consistent with the results obtained by Lauzier and colleagues, glucose starvation was sufficient to increase autophagic flux in HCT116 cells; however, this difference in protocols might potentially mask the effect of *CCNY* in autophagy. Moreover, HCT116 are colonic IECs, and defects in autophagy during IBD have been linked to Paneth cell dysfunction in the small intestine [41]. It may, thus, be interesting to repeat our experiments using cells that are more representative of Paneth cells, e.g., Caco-2.

Loss of cyclin Y could be compensated by redundancy with homologs, such as cyclin Y-like 1, which has overlapping functions, at least in Wnt signaling and mouse embryonic development [9,12]. However, within the intestinal epithelium, *CCNYL1* expression peaks in mature enterocytes, while *CCNY* is expressed mainly in stem cells and cycling transit amplifying cells, which are the primary Wnt-regulated cells. We, thus, consider it unlikely that cyclin Y-like 1 can substitute for cyclin Y, at least in intestinal Wnt signaling.

*CCNY* was identified as a candidate risk gene in IBD in several genome-wide association studies [3,4,5,6,7], and it has been suggested that candidate genes should be prioritized for functional studies, such as the current one [6]. However, the physiological relevance of the single-nucleotide polymorphisms (SNP) that were discovered in these studies has not been established. IBD is a multigenic disease [42], and thus, irrespective of the functional impact of these variants, additional genetic hits may be required for disease manifestation. Moreover, Franke et al. cautioned that, for several IBD risk loci, including those in the genomic region containing *CCNY*, it remains unclear which gene may be affected by specific SNPs [3]. Based on our functional experiments on cyclin Y biology in the gut, we propose that IBD-associated *CCNY* mutations—at least in isolation—are likely not critical for intestinal epithelial cell homeostasis.

## Figures and Tables

**Figure 1 cells-10-02330-f001:**
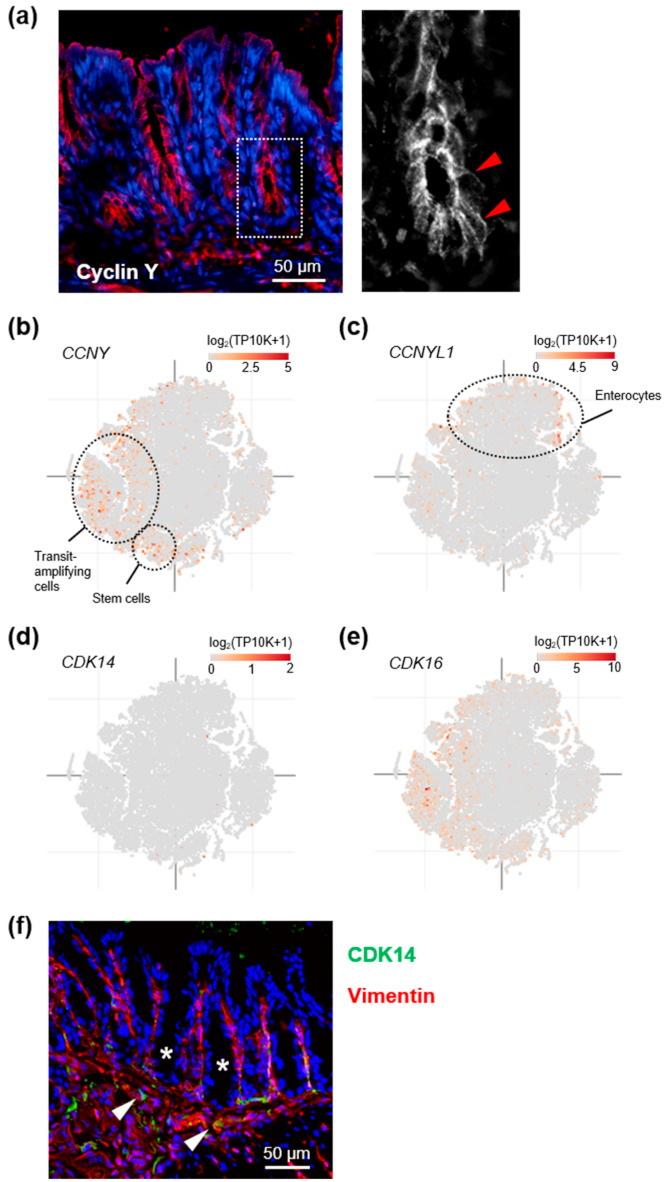
Cyclin Y, but not CDK14, is expressed in colonic epithelial cells. (**a**) Representative immunostaining of cyclin Y (red, with nuclei in blue) in the colon of wild-type mice. The magnified inset on the right highlights cyclin Y localization to the plasma membrane of crypt base intestinal epithelial cells (arrowheads). (**b**–**e**) Gene expression analysis of *CCNY* (**b**), *CCNYL1* (**c**), *CDK14* (**d**), and *CDK16* (**e**) in a single-cell RNA-seq dataset of intestinal epithelial cells from ulcerative colitis and control patients (Single Cell Portal accession SCP259). Peak expression of *CCNY* and *CDK16* was observed in epithelial stem and progenitor cells, whereas *CCNYL1* was more broadly expressed. Additional information on cell clusters can be found in Appendix A. (**f**) CDK14 staining (green, with nuclei in blue and vimentin in red) in wild-type mouse colon. CDK14 was detected in stromal cells (arrowheads), but not intestinal epithelial cells (*).

**Figure 2 cells-10-02330-f002:**
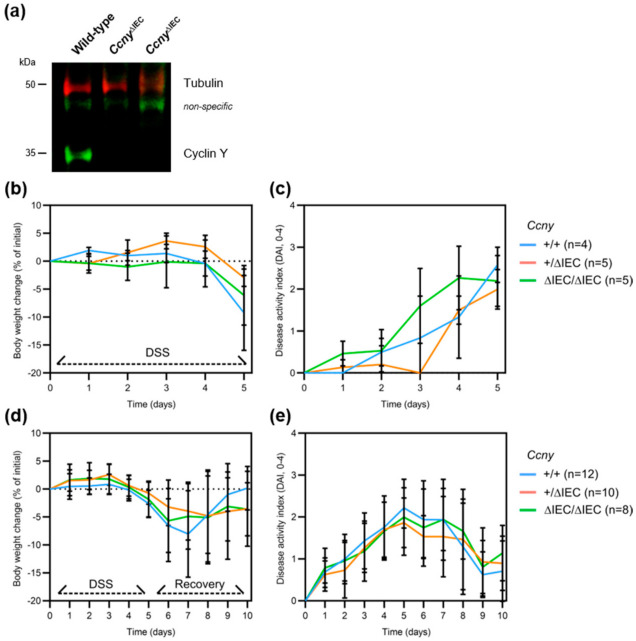
*Ccny*^Δ^^IEC^ mice do not display altered colitis susceptibility or injury repair. (**a**) Immunoblot validation of loss of cyclin Y (green) in colonic mucosal scrapings of two *Ccny*^Δ^^IEC^ mice and a wild-type littermate control. Tubulin (red) was used as a loading control. (**b**) Body weight change and (**c**) clinical disease activity following dextran sulfate sodium (DSS)-induced acute colitis (5 days). (**d**) Body weight change and (**e**) clinical disease activity during the recovery from DSS colitis (10 days). In these assays, *Ccny*^Δ^^IEC^ mice were indistinguishable from littermate controls.

**Figure 3 cells-10-02330-f003:**
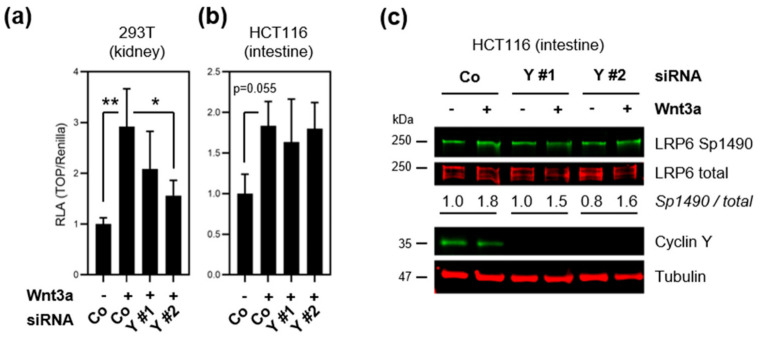
*CCNY* depletion does not affect Wnt signaling in intestinal epithelial cells. (**a**,**b**) Wnt activity upon cyclin Y loss-of-function by RNA interference with two siRNAs (Y #1/2) was assessed using the TOPflash luciferase reporter assay in 293T (**a**) and HCT116 cells (**b**). Where indicated, cells were treated with Wnt3a conditioned media. Data were analyzed by Dunnett’s post hoc test following one-way ANOVA, and statistically significant results are indicated as * *p* < 0.05 and ** *p* < 0.01 (versus siCo–Wnt3a). (**c**) Immunoblot analysis of active LRP6 (Sp1490) following cyclin Y loss-of-function and Wnt stimulation. The ratio of Sp1490 versus total LRP6 was normalized to the untreated siControl condition. Wnt-induced LRP6 phosphorylation was not affected by *CCNY* depletion.

**Figure 4 cells-10-02330-f004:**
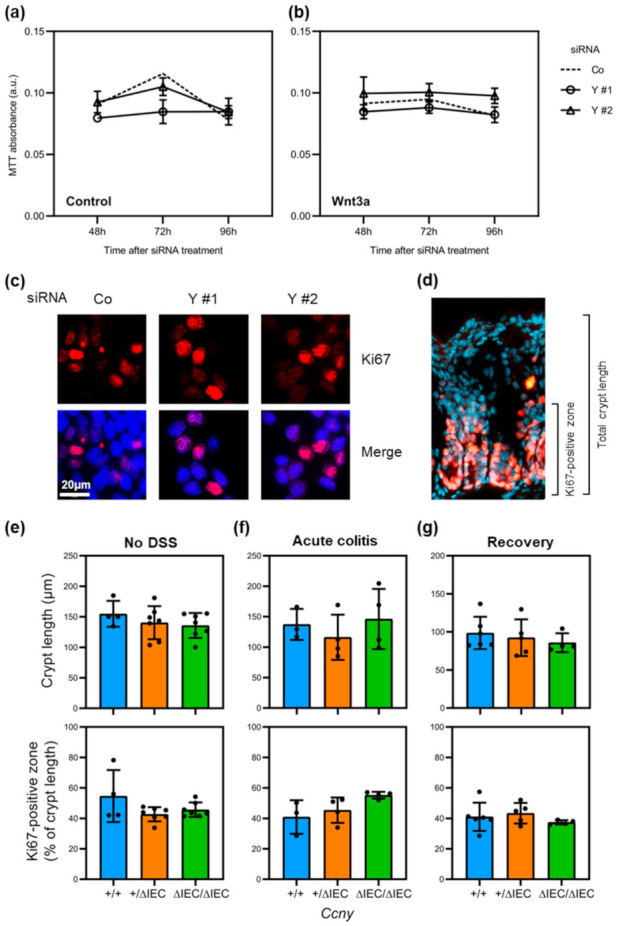
Loss of cyclin Y does not alter intestinal epithelial proliferation. (**a**,**b**) Cell proliferation of HCT116 cells upon cyclin Y loss-of-function by RNA interference was assessed with the MTT tetrazolium dye assay over a 96-h period. Wnt pathway activity was modulated by treatment with control conditioned media (**a**) or Wnt agonist Wnt3a (**b**). The dashed line indicates the average MTT absorbance of the siControl condition; a.u., arbitrary units. (**c**) Immunostaining of proliferation marker Ki67 (red, with nuclei in blue) in HCT116 cells following *CCNY* depletion for 48 h. (**d**) Representative immunostaining of Ki67 (red, with nuclei in blue) in the colon of a wild-type mouse. The Ki67-positive zone demarcates the crypt proliferative compartment. (**e**–**g**) Quantification of total crypt length and the crypt proliferative compartment in the distal colon as a fraction of the total crypt length in (**e**) unchallenged mice, (**f**) mice with acute DSS-induced colitis (5 days), and (**g**) mice following recovery from DSS colitis (10 days). Data points indicate individual animals. Only samples with a sufficiently high number of correctly aligned crypts were included in the analysis. No statistically significant results were observed between the groups (one-way ANOVA with Tukey’s post hoc comparison).

**Figure 5 cells-10-02330-f005:**
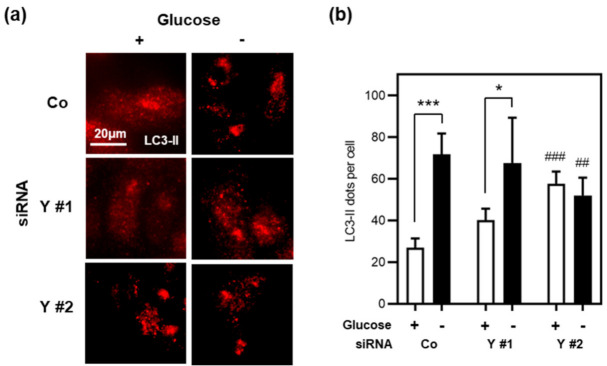
*CCNY* does not regulate autophagy in glucose-starved intestinal epithelial cells. (**a**) Immunofluorescence analysis of endogenous autophagosome marker LC3-II in HCT116 cells upon cyclin Y loss-of-function by RNA interference. Cells were maintained for 16 h in complete media or glucose starvation media with 100 nM Bafilomycin A1. (**b**) The average number of LC3 dots per cell in panel (**a**) was quantified by computer-assisted image analysis (*n* ≥ 6 images per condition, with ≥149 cells per image). Statistical analysis was performed using mixed-effects analysis (two-way ANOVA) followed by Dunnett’s post hoc test, and is presented as follows: * *p* < 0.05 and *** *p* < 0.001 versus complete media; ## *p* < 0.01 and ### *p* < 0.001 versus siControl.

## Data Availability

All newly generated data are contained within the article or supplementary material. Additionally, the following public gene expression datasets were analyzed: SCP259, GSE117993, and E-MTAB-2706.

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
