# Peer review of "The Candidate IBD Risk Gene CCNY Is Dispensable for Intestinal Epithelial Homeostasis"

_cells, 2021, doi:10.3390/cells10092330_

Round 1
Reviewer 1 Report
In this study, the authors aimed at analyzing a possible role of cyclin Y (cycY) in intestinal epithelial cells (IEC). Mice deficient in cycY expression in IEC were employed as well as an IEC-derived cell line in which cycY expression was targeted by siRNA. Lastly, a data bank search for the expression of the cycY afector CDK14 was performed. In summary, data are presented indicating that cycY does not affect experimentally induced colitis or recovery in mice, nor does it affect Wnt signaling, proliferation and autophagy in HCT116 cells. The authors conclude, that cycY is not critical for gut homeostasis.
The question risen is interesting and follows observations made in IBD patients, thus, has a potential clinical impact. The experiments are well designed, performed, and sufficiently documented. The discussion is adequate. Noteworthy, this manuscript presents ‘negative data’, a widely neglected area, which desperately deserves reanimated attention.
Although the manuscript is worth to be published, previously some revision need to be done
- In the abstract, analysis of ‘model IEC cell lines’ is indicated, while in the study only a single IEC-derived cell line (HCT116) was used. Moreover, in line 278 the authors state, that ‘the results suggest that cyclin Y is not required for activation of WNT signaling in IEC’, although the respective data were generated in HCT116 only. The use of a single cell line does not justify such a general conclusion. Either, a series of additional cell lines have to be analyzed or the statements have to be focused to HCT116 cells.
- HCT116 cells are colon carcinoma derived, which often bears genetic alterations affecting Wnt signaling. Such context is missing in the discussion.
- Genome wide association studies indicating CCNY as IBD risk locus have been performed in humans. Is it similar in mice? Thus, is it justified to use the mouse experimental system? In the same line, human HCT116 cells were used to support the data obtained in mice. Would a mouse IEC cell line (or mouse primary epithelial cells) give identical results? In summary, human and mouse experimental systems have to be compared and well balanced with regard to the question risen.
- CDK14 expression data obtained from data bank analyses should be confirmed in the applied experimental systems.
- In Fig. 3 a,b, statistics should be provided.
- In Fig. 5b, it seems that Y#1 and Y#2 gave different results; the authors however concluded a similar effect. Please clarify.
Author Response
In this study, the authors aimed at analyzing a possible role of cyclin Y (cycY) in intestinal epithelial cells (IEC). Mice deficient in cycY expression in IEC were employed as well as an IEC-derived cell line in which cycY expression was targeted by siRNA. Lastly, a data bank search for the expression of the cycY afector CDK14 was performed. In summary, data are presented indicating that cycY does not affect experimentally induced colitis or recovery in mice, nor does it affect Wnt signaling, proliferation and autophagy in HCT116 cells. The authors conclude, that cycY is not critical for gut homeostasis.
The question risen is interesting and follows observations made in IBD patients, thus, has a potential clinical impact. The experiments are well designed, performed, and sufficiently documented. The discussion is adequate. Noteworthy, this manuscript presents ‘negative data’, a widely neglected area, which desperately deserves reanimated attention.
Reply: Thank you for the positive and constructive comments!
Although the manuscript is worth to be published, previously some revision need to be done.
- In the abstract, analysis of ‘model IEC cell lines’ is indicated, while in the study only a single IEC-derived cell line (HCT116) was used. Moreover, in line 278 the authors state, that ‘the results suggest that cyclin Y is not required for activation of WNT signaling in IEC’, although the respective data were generated in HCT116 only. The use of a single cell line does not justify such a general conclusion. Either, a series of additional cell lines have to be analyzed or the statements have to be focused to HCT116 cells.
Reply: We have now repeated key experiments on Wnt pathway regulation and cell proliferation in additional CRC cell lines (SW48 and Caco-2), which have also recently been used to study LRP6 biology (PMID 34359960). However, results from these experiments were inconclusive. In Caco-2 cells, we achieved an average CCNY knock-down efficiency of just 19% and 39% using our two siRNAs, which did not allow us to draw any reliable conclusions. In SW48, knock-down efficiency was 85% and 69%, respectively. Here, however, we found that these cells do not respond to Wnt ligands, as judged by LRP6 phosphorylation and TOPflash reporter assays. A possible explanation for this discrepancy is that in contrast to SW48 and Caco-2, HCT116 cells have deleterious mutations in the Wnt pathway antagonists RNF43 and ZNRF3 (see DepMap portal of the Broad Institute at https://depmap.org/portal/), which should increase Wnt receptor levels and thereby Wnt responsiveness.
Given these observations, and our unsuccessful attempts to perform complementary ex vivo experiments (see our response to referee 2, comment 3), we have to limit the conclusions to our in vivo data and results from HCT116 at this time. We have now revised the manuscript to highlight this limitation.
- HCT116 cells are colon carcinoma derived, which often bears genetic alterations affecting Wnt signaling. Such context is missing in the discussion.
Reply: We have now discussed the potential impact of Wnt pathway mutations on page 11 of the revised manuscript. Importantly, despite activating beta-catenin mutations, HCT116 cells retain Wnt responsiveness and can therefore be used for functional studies such as the TOPflash reporter assay (PMID 24162018). Although this is not the case for the other cell lines we tested (see comment 1), the potential cyclin Y-dependent phosphorylation of LRP6 can still be assessed in these cells since the mutations occur downstream of the receptor.
- Genome wide association studies indicating CCNY as IBD risk locus have been performed in humans. Is it similar in mice? Thus, is it justified to use the mouse experimental system? In the same line, human HCT116 cells were used to support the data obtained in mice. Would a mouse IEC cell line (or mouse primary epithelial cells) give identical results? In summary, human and mouse experimental systems have to be compared and well balanced with regard to the question risen.
Reply: Based on earlier studies from our lab and others, the function of cyclin Y in Wnt signaling and cell proliferation (PMIDs 27203244, 26590424, 20059949) as well as autophagy (PMID 32098961) appears to be highly conserved across species. Indeed, Davidson et al. reported that cyclin Y controls LRP6 phosphorylation and Wnt signaling also in Drosophila and Xenopus (PMID 20059949). Accordingly, the cyclin Y protein is highly conserved as well, with >98% sequence identity between mice and man. Thus, we consider it valid to use mouse and human model systems interchangeably in studying basic cyclin Y biology.
Regarding the specific SNPs identified by GWAS, because of the different architecture of the Ccny gene locus in mice, it is not possible to study the impact of individual SNPs in mouse cells; however, this was not the purpose of the current study.
- CDK14 expression data obtained from data bank analyses should be confirmed in the applied experimental systems.
Reply: We have now tested the levels of CDK14 protein in different CRC cell lines as well as in mouse tissues. The results are shown in new Figures 1f and S1c. In summary, in agreement with the RNA data, we did not detect CDK14 protein in intestinal epithelial cells in vitro and in vivo.
- In Fig. 3 a,b, statistics should be provided.
Reply: As requested, we have analyzed the data in Figure 3 and added statistics.
- In Fig. 5b, it seems that Y#1 and Y#2 gave different results; the authors however concluded a similar effect. Please clarify.
Reply: In repeat experiments, we observed that the results in Figure 5 are fully reproducible, i.e. our CCNY siRNA #2 had a stronger effect on autophagy induction in non-starved cells compared to siRNA #1. This is indeed also the case for the TOPflash data in Figure 3A, where we observed a greater reduction in reporter activity in 293T cells in multiple independent experiments using siRNA #2.
We have carefully re-evaluated all available cyclin Y knock-down data, and found that siRNA #2 reduced cyclin Y levels slightly more efficiently across all experiments (see for example Figure S4a). Thus, we believe that the different results obtained with these siRNAs can be explained by their different efficiency. We have added a statement to this effect on page 8 of the revised manuscript.
Reviewer 2 Report
CCNY, encoding Cyclin Y, has been identified as a putative risk gene in IBD, including Crohn’s disease and ulcerative colitis, although the function of CCNY in the gut is unknown. Cyclin Y is known to be a critical activator of the Wnt/β-catenin signalling pathway, which controls stemness and proliferation in intestinal epithelia. The authors investigated whether CCNY regulates epithelial homeostasis and wound repair in the colon and contribute to epithelial regeneration following colitis, using in vivo and in vitro experiments. The results described far suggest that IBD risk gene CCNY is dispensable for intestinal epithelial homeostasis. Although data reported are not positive results, the findings are of interest. Experiments were carefully conducted. However, this reviewer would like to point the followings for publication.
- M & M: Experimental protocol of DSS-induced colitis should carefully be described for us to easily follow. For example, dosing period of DSS is unknown. Describe where the tissues were obtained from.
- Macroscopic and microscopic views of DSS-induced colitis should be provided.
- English needs editing.
Author Response
CCNY, encoding Cyclin Y, has been identified as a putative risk gene in IBD, including Crohn’s disease and ulcerative colitis, although the function of CCNY in the gut is unknown. Cyclin Y is known to be a critical activator of the Wnt/β-catenin signalling pathway, which controls stemness and proliferation in intestinal epithelia. The authors investigated whether CCNY regulates epithelial homeostasis and wound repair in the colon and contribute to epithelial regeneration following colitis, using in vivo and in vitro experiments. The results described far suggest that IBD risk gene CCNY is dispensable for intestinal epithelial homeostasis. Although data reported are not positive results, the findings are of interest. Experiments were carefully conducted. However, this reviewer would like to point the followings for publication.
Reply: Thank you for the positive and constructive comments.
- M & M: Experimental protocol of DSS-induced colitis should carefully be described for us to easily follow. For example, dosing period of DSS is unknown. Describe where the tissues were obtained from.
Reply: We have extensively revised the methods section of our manuscript to add the requested experimental details and well as additional information. Briefly, DSS was administered for 5 days, and analyses were performed on sections from the distal colon.
- Macroscopic and microscopic views of DSS-induced colitis should be provided.
Reply: Representative H&E stainings from acute DSS colitis and recovery can be found in new supplemental figure S2a-c. Unfortunately, the quality of the sections was insufficient for a detailed microscopic analysis of tissue damage. We are therefore including additional quantitative measures of colitis activity (body weight change, colon length, and colon weight/length ratio) as new supplemental figures S2d-f.
- English needs editing.
Reply: We have now carefully revised the manuscript to improve the language quality.
Reviewer 3 Report
This study investigated the contribution of cyclin Y (one of the risk genes for IBD) in the intestinal epithelium to maintaining intestinal homeostasis.
This study is fascinating because it used mice with KO of CCNY, specifically in intestinal epithelial cells.
This reviewer would like you to answer the following two questions.
1. Is there any difference in the intestinal microflora between KO mice and wild-type mice?
2. This study focuses on the role of CCNY in the colon. However, given the role of CCNY in gut homeostasis, the authors should examine the regeneration and proliferation of small intestinal mucosa in CCNY epithelial-specific KO mice.
3. In this study, we used a colorectal cancer cell line (HCT116). The authors should isolate intestinal epithelium from the mucosa of the small and large intestine of WT and KO mice and then prepare organoids for functional analysis.
Author Response
This study investigated the contribution of cyclin Y (one of the risk genes for IBD) in the intestinal epithelium to maintaining intestinal homeostasis.
This study is fascinating because it used mice with KO of CCNY, specifically in intestinal epithelial cells.
Reply: Thank you!
This reviewer would like you to answer the following two questions.
- Is there any difference in the intestinal microflora between KO mice and wild-type mice?
Reply: In our experiments, we did not collect stool samples that would allow us to address this question experimentally. However, in the absence of an intestinal phenotype in Ccny cKO mice, we expect any potential differences to be subtle and of limited consequence for host physiology. Notably, global Ccny KO mice display an altered metabolism that may be compatible with changes in the gut microbiome (PMID 26161966). Since this phenotype does not manifest upon loss of Ccny in intestinal epithelia only, we consider it unlikely that epithelial cyclin Y is involved in the regulation of the intestinal microflora.
- This study focuses on the role of CCNY in the colon. However, given the role of CCNY in gut homeostasis, the authors should examine the regeneration and proliferation of small intestinal mucosa in CCNY epithelial-specific KO mice.
Reply: While it would be interesting to study the role of cyclin Y in small intestinal regeneration as well, to our knowledge there are no suitable (non-genetic) enteritis models to address this question in vivo. Indeed, in the DSS colitis model used here, epithelial injury and inflammation are largely restricted to the descending colon. It may be possible to further explore the role of cyclin Y in (non-inflammatory) injury repair using small intestinal organoids; however, as detailed in the response to the following comment, we were unable to perform these functional assays in time for this revision.
Regarding proliferation, when we initially characterized our Ccny cKO mice, we observed no obvious differences in small intestinal histomorphology during normal homeostasis or DSS colitis, which argues against altered IEC proliferation. Importantly, this finding is fully consistent with a recently published study by Raisch et al. (PMID 34359960), who observed no changes in IEC proliferation, histology, and stem cell numbers in the jejunum of Lrp6 cKO mice. We have now added this information and reference on page 7 of the revised manuscript.
- In this study, we used a colorectal cancer cell line (HCT116). The authors should isolate intestinal epithelium from the mucosa of the small and large intestine of WT and KO mice and then prepare organoids for functional analysis.
Reply: Unfortunately, despite our best efforts we were unable to establish enteroids from our cKO mice in the time-frame of this revision. We are currently attempting loss-of-function studies in small intestinal wild-type organoids, but have not achieved sufficient Ccny knock-down for meaningful functional studies. Moreover, we discovered that other commonly used CRC cell lines do not respond to Wnt ligands, and are therefore not suited for studying cyclin Y biology (see our response to referee 1, comment 1 for more details). Especially in light of the recent observation that Lrp6-deficient enteroids exhibit delayed growth (PMID 34359960), we agree that it would be very interesting to further investigate cyclin Y biology using this model system. However, at this point we respectfully propose to address this question in a follow-up study, and hope that the complementary in vivo and in vitro data we present in this study sufficiently support our conclusions.
Round 2
Reviewer 1 Report
All my comments have been addressed adequately.
Reviewer 2 Report
The revised manuscript has greatly been improved and reply to the comments is adequate. It is now acceptable in the journal.
Reviewer 3 Report
This reviewer understands that it is difficult to conduct the additional experiments at this time.